# A Graph Theoretic Framework of Recomputation Algorithms for Memory-Efficient Backpropagation

**Mitsuru Kusumoto** *
Preferred Networks, Inc.
mkusumoto@preferred.jp

**Takuya Inoue** *
The University of Tokyo
inoue-takuya57@g.ecc.u-tokyo.ac.jp

**Gentaro Watanabe**
Preferred Networks, Inc.
g.wtnb@preferred.jp

**Takuya Akiba**
Preferred Networks, Inc.
akiba@preferred.jp

**Masanori Koyama**
Preferred Networks, Inc.
masomatics@preferred.jp

## Abstract

*Recomputation algorithms* collectively refer to a family of methods that aims to reduce the memory consumption of the backpropagation by selectively discarding the intermediate results of the forward propagation and recomputing the discarded results as needed. In this paper, we will propose a novel and efficient recomputation method that can be applied to a wider range of neural nets than previous methods. We use the language of graph theory to formalize the *general recomputation problem* of minimizing the computational overhead under a fixed memory budget constraint, and provide a dynamic programming solution to the problem. Our method can reduce the peak memory consumption on various benchmark networks by $36\% \sim 81\%$, which outperforms the reduction achieved by other methods.

## 1 Introduction

The efficiency of memory usage is always one of the most important issues in the application of deep neural nets. Modern deep neural networks used in commercial applications tend to be large, and they require massive memory for the forward and backward computations. The inputs to the networks can be large as well; this is particularly true for the tasks related to computer vision such as object detection and semantic segmentation, where higher resolution images are generally more useful to detect small objects accurately. Ample free memory is also important for the training process itself; when the memory is insufficient, the user has no choice but to choose a small batch size. This is problematic, especially when using batch normalization [7] as the quality of batch statistics degrades. Indeed, its impact on the resulting model quality is crucial. Recent studies enabled themselves to use large batch sizes by reducing the memory consumption or introducing distributed computation, and succeeded in achieving the state-of-the-art results for computer vision tasks [12, 20].

*Recomputation algorithms* collectively refer to a family of methods of smart memory manipulation that can reduce the memory consumption without altering the outcome of the computation or compromising the accuracy of the output. In theory, backward computation requires the result of the forward computation, and naive approach requires the user to cache all the forward computation results for the backward computation. However, we can reduce the peak memory consumption by deliberately discarding some parts of the intermediate results in the forward computation, and *recompute* these intermediate results gradually on a need basis during the backpropagation. Naturally, the efficacy of any recomputation method depends on its rules of *what to forget and what to cache in what order*.

---

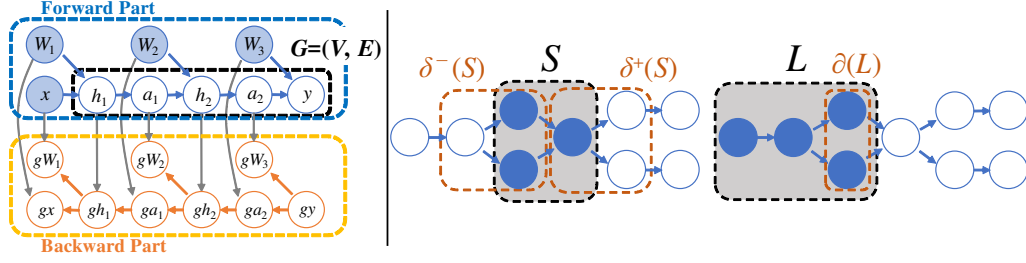

Figure 1: (Left) The computation graph of a three-layer perceptron. A part induced by intermediate nodes is denoted by $G = (V, E)$. (Right) Neighborhoods, lower set, and its boundary.

Indeed, the idea of recomputation is not entirely new on its own. For example, Chen et al. [2] have proposed an efficient recomputation framework for a specific family of networks that can be divided into *segments*, and Gruslys et al. [4] have proposed a method specializing in RNNs. These methods, however, do not investigate neural networks with complex structures in complete generality, and their applications are limited either to a specific family of networks or to a group of networks to which their ad-hoc heuristics can be applied.

In this paper, we will propose a novel and efficient recomputation method that can be applied to theoretically all types of neural nets. To the authors' knowledge, there has been no study to date that tackled this problem in a completely general setting, and a new formulation of the recomputation problem is warranted. We will, therefore, define the *general recomputation problem* with appropriate formality as minimization of computational overhead under a fixed memory budget (Section 2 and 3), and provide our dynamic programming (DP) solution to the problem (Section 4.1 to 4.3). We will also show that, by solving the opposite problem of maximizing the computational overhead, we can reduce the peak memory consumption to the level that cannot be achieved by contemporary methods (Section 4.4). We will demonstrate the efficacy of our DP solution on the various benchmark networks (ResNet, DenseNet, U-Net, PSPNet, etc.) and compare its performance against contemporary methods in terms of the computational overhead and the size of the achieved memory reduction (Section 5). Our method can reduce the peak memory consumption by $36\% \sim 81\%$, which is much greater than the reduction that can be achieved by other methods.

## 2 Preliminaries

In this first section, we present basic concepts and definitions that will be required for understanding our algorithms. Throughout, we will use a directed graph $G = (V, E)$ to represent the architecture of the network, where $V$ is the set of variables and there is an edge $(v, w) \in E$ if $v \in V$ is directly required for the computation of $w \in V$. In our study, we will exclude the *input nodes* from the definition of $V$, because the computations on the *intermediate nodes* tend to play a more pivotal role than the input nodes for the memory consumption of neural networks. We also exclude from our considerations the memory required for the model parameters as well. In computation theory, it is customary to use *computational graph* to represent the computational dependency among the variables. Figure 1-Left is the computational graph for the forward and backward computation on a neural network. Although we do not explicitly use the computational graph in our formulation of the problem, we will develop our algorithm by building a theory on the computational graph.

For an arbitrary node set $S \subseteq V$, we will use $\delta^+(S)$ to denote the set of nodes to which there is a directed edge from some node in $S$. We define $\delta^-(S)$ analogously: $\delta^+(S) := \{v \in V \mid (s, v) \in E \text{ for some } s \in S\}$ and $\delta^-(S) := \{v \in V \mid (v, s) \in E \text{ for some } s \in S\}$.

Also, following the definitions in order theory [3], we say $L \subseteq V$ is a *lower set* of $V$ if there is no edge from $V \setminus L$ to $L$, and write $L \prec V$. By definition, $L$ is a lower set if and only if $\delta^-(L) \subseteq L$. The boundary of $L$ is defined by $\partial(L) := \delta^-(V \setminus L) \cap L$. In our theory, the concept of the lower set we introduced above plays a pivotal role. We denote the set of all lower sets by $\mathcal{L}_G$. By a simple argument, we can deduce that $\#V \leq \#\mathcal{L}_G \leq 2^{\#V}$ holds for any graph $G$. See Figure 1-Right for the visual renditions of these definitions.

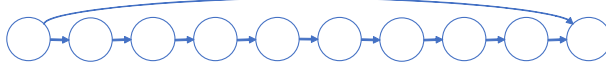

Figure 2: A network that cannot be divided into multiple segments.

Finally, we also define the forward computation cost $T_v > 0$ and the memory consumption cost $M_v > 0$ to each node $v \in V$. Likewise, we define $T(S) := \sum_{v \in S} T_v$ and $M(S) := \sum_{v \in S} M_v$ for $S \subseteq V$. We do not need to consider the backward computation cost in this study, because we will not recompute the backward computation.

**Chen's Recomputation Algorithm**   Before explaining our formulation, we would like to introduce the work of Chen et al. [2] in order to provide some intuition for our approach. Chen et al. proposed a method of recomputation for a family of neural nets that can be decomposed of *segments*. For an $n$ layer network, Chen's algorithm divides the graph into $\sqrt{n}$ segments of length $\sqrt{n}$. In the forward computation, the algorithm caches the values for the nodes at the boundary of each segment and discard all other nodes from the memory. When the time comes for backpropagation of the segment $i$, it recomputes the required forward propagation results using the cache of the segment $i-1$. This way, the algorithm can keep the peak memory consumption within $O(\sqrt{n})$ at an additional computational cost that amounts to one round of the forward computation. As mentioned in the beginning, however, their algorithm can be applied only to a specific type of graph.

For example, a neural net with a skip connection that connects the input layer with the output layer cannot (Figure 2) be divided into more than one segment, and an extra set of heuristical rules have to be applied in order to deal with such a situation. In general, it is difficult to determine if a given arbitrary graph can be handled by their heuristics.

## 3   General Recomputation Problem

In this study, we will extend Chen's framework to a generic graph by reformulating the recomputation problem so that we can develop algorithms that can be applied to all types of graphs. We first need an additional set of definitions and notations.

As mentioned in the introduction, the efficacy of any recomputation method is determined by its rules of caching and its order of computation. Consider a partition $V = \bigcup_{i=1}^{k} V_i$ with the intention of computing $V_i$ after $V_{i-1}$ in the forward computation. In order to make computation in the intended order, we need to require that if $(v, w) \in E$, then $v \in V_i$ and $w \in V_j$ must hold for some $i \leq j$. If this requirement is satisfied for the partition, we can construct an increasing sequence of lower sets $\{L_1 \prec L_2 \prec \ldots \prec L_k = V\}$ by putting $L_i := V_1 \cup V_2 \cup \ldots \cup V_i$. Indeed, we can do the construction backward by starting from an arbitrary increasing sequence $\{L_1 \prec L_2 \prec \ldots \prec L_k = V\}$ and defining $V_i = L_i \setminus L_{i-1}$ (Figure 3-(a)). These building blocks will be the basis of our formulation.

Now, given an arbitrary sequence of lower sets $\{L_1 \prec \ldots \prec L_k = V\}$, we define the *canonical strategy* of recomputation as follows:

**Forward computation**   After making the evaluations for all nodes in $V_i$, cache the values associated with $\partial(L_i)$ and discard all the values for $V_i \setminus \partial(L_i)$. Using the cache of $\partial(L_i)$, compute the values for $V_{i+1}$.

See Figure 3-(b) for a visualization of the forward computation. At the time of completion of the computations for $V_i$, the nodes with solid blue color have been cached, and the nodes with opaque blue colors have been discarded.

**Backward computation**   Backward computations are to be conducted in the reverse order. After making the evaluations for all nodes in $V_{i+1}$ (Figure 3-(c)), we will execute the following commands in order.

1. recover the required forward values for the nodes $V_i$ based on the cache of $\partial(L_{i-1})$ and conduct the backpropagation for the nodes in $V_i$. (Figure 3-(d))
2. cache the nodes in the backward part of the computational graph that corresponds to $\delta^+(L_{i-1}) \cap V_i$, and discard the nodes in both the forward and backward part that correspond to the nodes of $V_i$ that **will not be needed in the computation in future.**

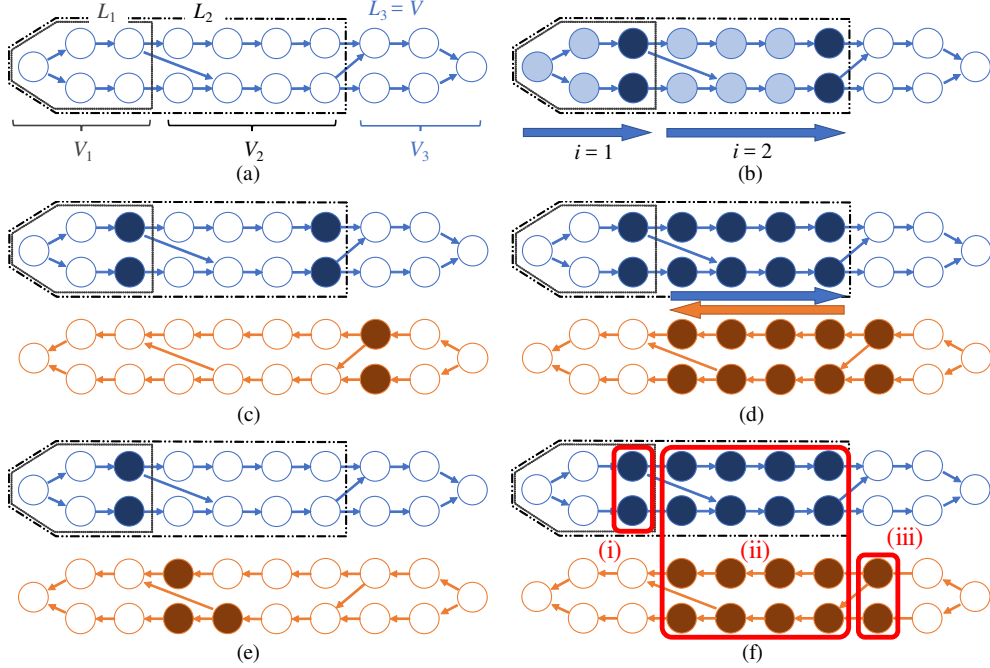

Figure 3: The visualization of the canonical strategy for a lower set sequence.

This means that if there is a skip connection into $v \in V_i$, the cache on $v$ will not be discarded. (Figure 3-(e))

In the real implementation of this canonical strategy, the gradient of the model with respect to the parameters are to be reported to the user in real time during the process of backpropagation.

In principle, our formulation is based on this canonical strategy. Chen's algorithm can be seen as an instance of the canonical strategy that specializes in a specific form of lower sets.

Now, the efficacy of this canonical strategy will depend solely on the choice of the sequence of the lower sets. Two performance measures are of our interest: the amount of computational overhead and the size of peak memory consumption. First, at the end of the forward computation for $V_i$, the values for the following nodes $U_i$ have been cached by the algorithm: $U_i := \bigcup_{j=1}^{i} \partial(L_j)$.

Thus, the set of nodes subject to the recomputation is given by $V \setminus U_k$. The total computational overhead of the recomputation is given by

$$T(V \setminus U_k) = \sum_{i=1}^{k} T\Big(V_i \setminus \partial(L_i)\Big). \tag{1}$$

We would like to minimize this value under the prescribed memory budget. Let us also denote the computational overhead for the strategy based on the same sequence by $T(\{L_1 \prec \ldots \prec L_k\})$.

By the design of the canonical strategy, the memory consumption shall reach its peak during the backward computation. The breakdown of the memory consumption of the backward computation for $V_i$ is as follows (Figure 3-(f)):

   (i) Cache for the forward computations up to $V_i$ requires $M(U_{i-1})$.
  (ii) Altogether, the forward part and the backward part for $V_i$ requires $2M(V_i)$.
 (iii) In backpropagation, we need to take into account the set of nodes in $V_j$ with $j > i$ that are dependent on the nodes in $V_i$. Such nodes require $M(\delta^+(L_i) \setminus L_i)$.
 (iv) When there are edges from $v_1, v_2, v_3$ to $h$, one might have to use the forward cache dedicated to $v_1$ and $v_2$ for the gradient at $v_3$. Such case will require $M(\delta^-(\delta^+(L_i)) \setminus L_i)$.

In total, the backpropagation for the nodes in $V_i$ requires

$$\mathcal{M}^{(i)} := M(U_{i-1}) + 2M(V_i) + M(\delta^+(L_i) \setminus L_i) + M(\delta^-(\delta^+(L_i)) \setminus L_i). \qquad (2)$$

The peak memory consumption of the canonical strategy is therefore given by $\max_{i=1,\dots,k} \mathcal{M}^{(i)}$. Again, this is a value that is solely determined by the sequence of the lower sets used in its evaluation. Let us, therefore, use the notation $M(\{L_1 \prec \dots \prec L_k\})$ to denote the peak memory consumption of the canonical strategy based on $\{L_1 \prec \dots \prec L_k = V\}$. We can formalize our problem as follows:

**Definition** (General Recomputation Problem). Given a neural network with graph representation $G = (V, E)$ and a prescribed memory budget $B$, find the increasing sequence $\{L_1 \prec \dots \prec L_k = V\}$ of lower sets that minimizes $T(\{L_1 \prec \dots \prec L_k\})$ while satisfying $M(\{L_1 \prec \dots \prec L_k\}) \leq B$. If $B$ is too small, there can be no solution to this problem.

After solving the general recomputation problem, one can aim to further improve the efficiency by executing the obtained strategy with popular heuristics like liveness analysis [1].

In order to solve the general recompuation problem in practice, we need a relative value of $T_v$ for each node $v$ in the network. We can either directly measure $T_v$ with some time scale or use some form of approximation. In general, convolutional node tends to be heavier than other types of node in terms of computational cost. In our experiment, we therefore set $T_v = 10$ for convolutional node, and $T_v = 1$ for all other types of node.

# 4 Methods

In this section, we will provide three ways to solve the general recomputation problem: (1) a naive exhaustive search, (2) an exact dynamic programming (DP) algorithm, and (3) an approximate DP algorithm that can be used to obtain a near-optimal canocnical strategy *fast*. We also present a *memotry-centric* strategy, which prioritizes the reduction of peak memory consumption over the computational overhead.

## 4.1 Naive Approach: An Exhaustive Search

Let us first consider a rudimentary exhaustive search algorithm based on depth-first search (DFS). Because the number of all lower sets $\#\mathcal{L}_G$ is finite, one can find a solution to the general recomputation problem in time of order $O(\#\mathcal{L}_G^{\#V})$ by simply computing $M(\{L_1 \prec \dots \prec L_k\})$ and $T(\{L_1 \prec \dots L_k\})$ for every sequence $\{L_1 \prec \dots \prec L_k = V\}$. For further discussion, let us extend the definition of $M(\{L_1 \prec \dots \prec L_k\})$ for the prefix of the lower set sequence using Equation (2). For $i \leq k$, let $M(\{L_1 \prec \dots \prec L_i\}) := \max_{j=1,\dots,i} \mathcal{M}^{(j)}$. We can similarly define the computational overhead using Equation (1). Let $T(\{L_1 \prec \dots \prec L_i\}) := \sum_{j=1}^{i} T(V_j \setminus \partial(L_j))$.

Starting from an arbitrary lower set $L_1$, the DFS proceeds by recursively visiting $\{L_1 \prec \dots \prec L_i\}$ from $\{L_1 \prec \dots \prec L_{i-1}\}$ such that $M(\{L_1 \prec \dots \prec L_i\}) \leq B$. From the definition above, we can deduce $T(\{L_1 \prec \dots \prec L_i\}) = T(\{L_1 \prec \dots \prec L_{i-1}\}) + T(V_i \setminus \partial(L_i))$. Thus, we can compute the computational overhead incrementally. The memory consumption can be computed incrementally as well. In Equation (2), the only term that depends on the entire sequence $\{L_1 \prec \dots \prec L_i\}$ is $M(U_{i-1})$, and all other terms can be computed directly from $L_i$ and $V_i = L_i \setminus L_{i-1}$.

A keen reader might have noticed at this point that there is no need to retain the information of the entire traversed path $\{L_1 \prec \dots \prec L_i\}$ in this DFS algorithm. Instead, it suffices to keep track of the triplets $(L, t, m)$ where $L = L_i$, $t = T(\{L_1 \prec \dots \prec L_i\})$, and $m = M(U_i)$.

## 4.2 Exact DP Algorithm

We can use the triplet representation $(L, t, m)$ used in the previous subsection to solve the general recomputation problem with DP. Algorithm 1 in the Appendix summarizes our DP procedure.

Let $\mathcal{S}$ be the set of all triplet states that can be visited during the DFS previously mentioned. Our DP solution takes advantage of the fact that, if there exist $(L, t, m)$ and $(L, t, m') \in \mathcal{S}$ such that $m < m'$, we do not need to consider $(L, t, m')$ for further exploration.

Let us define an array `opt` with entries $\texttt{opt}[L,t] := \min\{m \in \mathbb{N} \mid (L,t,m) \in \mathcal{S}\}$, where $\texttt{opt}[L,t] := \infty$ whenever there is no $m$ such that $(L,t,m) \in \mathcal{S}$. This will serve as our DP table. Starting with the initial condition $\texttt{opt}[\emptyset,0] = 0$, we fill the DP table by visiting the states that satisfy the prescribed memory budget constraint. More precisely, if $\texttt{opt}[L,t]$ is the current entry, the algorithm loops over the set of all $L'$ such that $L \subsetneq L'$ and update $\texttt{opt}[L',t']$ with $t' = t + T(V' \setminus \partial(L'))$.

If $\texttt{opt}[L,t] < \infty$ after its evaluation, it means that there exists $\{L_1 \prec \ldots \prec L_i\}$ with $L_i = L$ such that $M(\{L_1 \prec \ldots \prec L_i\}) \leq B$ and $T(\{L_1 \prec \ldots \prec L_i\}) = t$. If $t^* := \min\{t_0 \mid \texttt{opt}[V,t_0] < \infty\} < \infty$, we can report $t^*$ as the minimum possible computational overhead in the general recomputation problem. Conversely, if $t^* = \infty$, it means that there is no solution.

For the sake of argument, let us assume for now that $t^* < \infty$. In order to obtain the choice of $\{L_1 \prec \ldots \prec L_k = V\}$ that achieves $t^*$, we need modify the algorithm slightly by making a room for another variable $\texttt{optarg}[L',t']$ that stores the pointer to the $L$ that precedes $L'$ in the sequence of the recursive computation that leads to $\texttt{opt}[L',t']$. The optimal sequence of the lower sets can thus be obtained by starting from $\texttt{opt}[V,t^*]$ and tracing back the pointers of $\texttt{optarg}$.

Since the bottleneck of our DP algorithm is its iteration process, the computational time of our algorithm is $O(T(V) \cdot \#\mathcal{L}_G{}^2)$. From a practical perspective, it is worth noting that the use of a sparse table for `opt` reduces the computation time by a large constant factor. Further, when $t < t'$ and $\texttt{opt}[L,t] < \texttt{opt}[L,t']$, we can skip the iteration for the entry $\texttt{opt}[L,t']$.

## 4.3  Approximate DP Algorithm

The DP algorithm we described in the previous section can be used to obtain the optimal canonical strategy. However, the computation time required for the DP algorithm grows with $\#\mathcal{L}_G$, which can be very large for models with complex network structures. We also present an algorithm that can be used to obtain a near-optimal canonical strategy fast. We shall emphasize here that any canonical strategy is a legitimate recomputation strategy in the sense that it never alters the network output.

The modification we would make to the DP algorithm is simple. Instead of making keys for all members of $\mathcal{L}_G$ in the DP table, we use a good small subset of $\mathcal{L}_G$ at every cell.

Let us say that $v$ is *reachable* from $w$ if $v = w$ or if there exists a path from $w$ to $v$. Now, let us define $\mathcal{L}_G^{\text{Pruned}} := \{L^v \mid v \in V\}$ where $L^v := \{w \in V \mid v \text{ is reachable from } w\}$. By definition, $\mathcal{L}_G^{\text{Pruned}} \subseteq \mathcal{L}_G$ and $\#\mathcal{L}_G^{\text{Pruned}} = \#V$. Our approximate DP algorithm makes keys for the members of $\mathcal{L}_G^{\text{Pruned}}$ only. This way, we can keep the computation time under $O(T(V) \cdot \#V^2)$.

Indeed, this modification excludes some possibilities from the search pool, and we can no longer guarantee that we will be able to find the best canonical strategy. As we will show in our experiments, however, near-optimal solutions obtained from our approximate DP are often "good enough" in practice at reducing the peak memory consumption.

## 4.4  Memory-Centric Strategy

Liveness analysis [1] is a heuristic technique that has an effect on the reduction of peak memory consumption, and much of a given strategy's performance in terms of memory consumption depends on how well this technique works in the corresponding sequence of lower sets. Experimentally, liveness analysis tends to work well when the node-set is partitioned coarsely; that is, when each $V_i$ is large. Through trial and error, we discovered that we can realize this *coarse partition* intentionally by using a strategy with long computational overhead. In fact, a canonical strategy with maximal computational overhead tends to have exceptionally low peak memory consumption. Given a fixed budget constraint $B$, the canonical strategy with maximal computational overhead can again be found using DP.

In general, we can find a more economical strategy by setting the budget constraint $B$ to a smaller value. If the reduction of the memory consumption is the first priority, one may set $B$ to the lowest value for which the set of canonical strategies is non-empty. We call the strategy found this way a *memory-centric strategy*, because it prioritizes the positive effect of liveness analysis over the computational overhead. The computational overhead of memory-centric strategy is bounded by the

computation time for one round of the forward computation. We discriminate this strategy from the optimal canonical strategy in the previous section by calling the latter a *time-centric strategy*.

When applying our methods, we recommend the user to first try the time-centric strategy and prioritize the computational overhead. We suggest the user to try the memory-centric strategy and pursue memory reduction only if the solution of the time-centric strategy fails to satisfy the memory budget constraint even with the application of liveness analysis.

## 5 Experiments

We applied our algorithm to various network structures and investigated their performance in terms of computational overhead and peak memory consumption. All networks were implemented in Chainer [18], and the experiments were conducted on NVIDIA Tesla K40c with GPU DRAM of size 11.4 GB. The following are the list of networks on which applied our method: ResNet [5], VGG [16], DenseNet [6], GoogLeNet [17], U-Net [15], and PSPNet [20]. Input dimensions were set to $572 \times 572$ for U-Net, $713 \times 713$ for PSPNet, and $224 \times 224$ for all other networks.

We compare our method against two methods: (1) vanilla implementation of the forward and backward propagation without any recomputation methods, and (2) Chen's algorithm implemented with the use of their heuristic techniques. We tested both our method and Chen's method with the liveness analysis. In the Appendix, we present an ablation study for the effect of liveness analysis.

Our code is publicly available at `https://github.com/pfnet-research/recompute`.

### 5.1 Memory Reduction

Table 1 summarizes the performance of various methods evaluated in terms of the size of the achieved memory consumption. ExactDP and ApproxDP are algorithms in Section 4.2 and 4.3, respectively. MC stands for memory-centric strategy in Section 4.4 and TC stands for time-centric strategy. The peak memory consumption enlisted in this table includes the memory used by the model parameters itself. Each value inside the parenthesis designates the proportional size of the achieved memory reduction relative to the peak memory consumption of the vanilla run. For each experiment, we selected a batch size so that the memory consumption with vanilla run roughly falls in the range of $7 \sim 9$ GB. For the memory budget $B$ to be used for our approach, we chose the minimal value $B$ for which the solution of the general recomputation problem exists. This value was determined using binary search.

As we can confirm on the table, our method outperforms the previous method in terms of the peak memory consumption. In DenseNet and PSPNet, we are succeeding in reducing the memory consumption respectively by 81% and 71%. Our method is performing better than Chen's algorithm particularly for complex networks like PSPNet, U-Net, and GoogLeNet. The approximate DP was siginificantly faster to complete than the exact DP algorithm. The exact DP algorithm required more than 80 secs to complete for GoogLeNet and PSPNet, while the approximate DP completed within 1 sec for all networks. We would like to emphasize that, for all cases we considered, the exact solution and approximate solution did not differ much in terms of performance. This is to say that the "near-optimal" canonical strategy obtained from the Approximate DP is literally "near" optimal for complex networks commonly used in applications, and that it can be used reliably in practice. [2]

### 5.2 Computational Time

We investigated the memory-runtime tradeoff for our algorithm by running the experiments with various batch sizes and compared the results against other methods. For all the experiments of recomputation methods, we used batch sizes that are so large that the naive vanilla computation is impossible. For each choice of batch size, we repeated each experiment four times and reported their average runtime.

Table 1: The comparison of peak memory consumption. Each value inside the parenthesis is the achieved memory reduction from the vanilla computation.

| Network | ApproxDP + MC | ApproxDP + TC | ExactDP + MC | ExactDP + TC | Chen's [2] | Vanilla | #V | Batch |
|---------|---------------|---------------|--------------|--------------|------------|---------|-----|-------|
| PSPNet | **2.7 GB (-71%)** | 3.1 GB (-67%) | 2.8 GB (-70%) | 3.2 GB (-66%) | 4.0 GB (-58%) | 9.4 GB | 385 | 2 |
| U-Net | 5.0 GB (-45%) | 6.7 GB (-26%) | **4.7 GB (-48%)** | 5.3 GB (-42%) | 7.4 GB (-18%) | 9.1 GB | 60 | 8 |
| ResNet50 | **3.4 GB (-62%)** | 4.4 GB (-51%) | **3.4 GB (-62%)** | 4.3 GB (-51%) | 3.7 GB (-59%) | 8.9 GB | 176 | 96 |
| ResNet152 | **2.3 GB (-75%)** | 2.5 GB (-73%) | **2.3 GB (-75%)** | 2.5 GB (-73%) | 2.4 GB (-74%) | 9.2 GB | 516 | 48 |
| VGG19 | **4.5 GB (-36%)** | 5.5 GB (-22%) | **4.5 GB (-36%)** | 5.5 GB (-22%) | 4.7 GB (-34%) | 7.0 GB | 46 | 64 |
| DenseNet161 | **1.6 GB (-81%)** | 1.9 GB (-78%) | 1.7 GB (-80%) | 1.8 GB (-78%) | 1.8 GB (-79%) | 8.5 GB | 568 | 32 |
| GoogLeNet | **5.2 GB (-39%)** | 5.5 GB (-36%) | **5.2 GB (-39%)** | 5.9 GB (-31%) | 6.5 GB (-24%) | 8.5 GB | 134 | 256 |

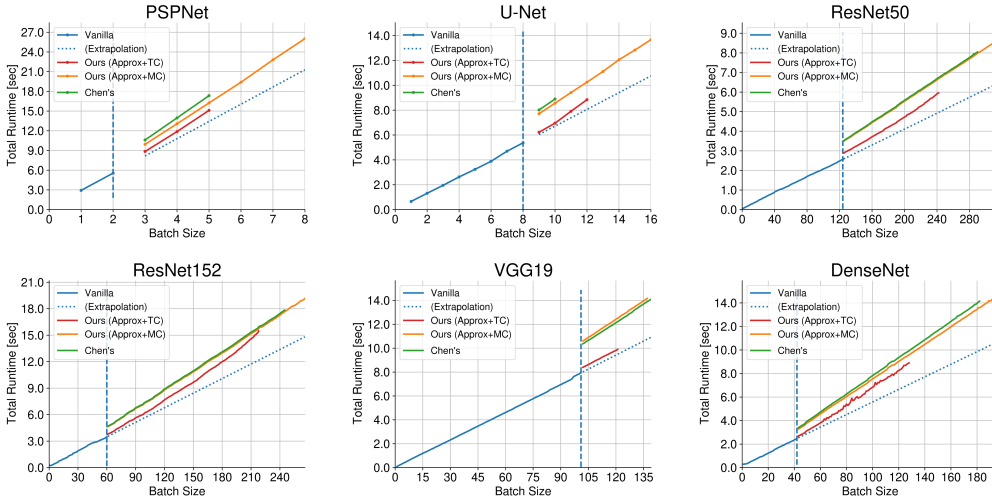

Figure 4: The tradeoff between batch size and total runtime (forward and backward propagation).

Figure 4 illustrates the results of this set of experiments. Blue curves are the results obtained from the vanilla computation. Dotted blue lines are the linear extrapolation of the vanilla translation results that one might have been able to obtain if there was more memory in the device. Red curves and orange curves respectively designate the results of "ApproxDP + time-centric" and "ApproxDP + memory-centric" settings. Green curves represent the method obtained by Chen's algorithm. As we can see in the plot, our method outperforms Chen's method by a large margin in terms of the runtime-memory tradeoff. In terms of the runtime that will be required to conduct an experiment on ResNet152 with the batch size that is double the maximum batch size that can be used in vanilla computation, our method was 1.16 times faster than Chen's algorithm. This results also verify that our algorithm indeed seeks a strategy with small computation overhead in presence of ample memory. For PSPNet, our method was able to increase the maximum possible batch size from 2 to 8.

## 6 Related Work

There are various other methods of reducing the memory consumption that do not belong to the family of recomputation methods. Precision reduction [11] reduces the memory consumption by regulating the precision level of the computations. Network pruning [10], on the other hand, takes the approach of compressing the network itself. These methods are fundamentally different from recomputation methods in that they compromise the accuracy of the output in favor of efficient memory usage, because the name *recomputation* refers to a family of methods that does not alter the output of the network in anyway whatsoever. At the same time, however, the methods we mentioned above can be used in combination with recomputation methods if the reduction of the peak memory consumption is the first priority. Another method is to selectively transfer some part of the memory between CPU and GPU during the training process. Superneurons [19], vDNN [14], and LayRub [8] are the variants of this approach. This method, too, can be used in combination with recomputation methods.

Some other methods specialize in the memory consumption reduction for a specific family of networks. Pleiss et al. [13] developed a method that specializes in DenseNet by storing multiple feature maps at a shared memory location. Gruslys et al. [4] developed a method that specializes in RNNs.

Chen et al. [2] developed a recomputation framework for a family of graphs that can be divided into segments and provided a set of heuristic rules that can be used to extend the framework to select networks like LSTM and ResNet. Our framework is more general and powerful than Chen's framework. Chen's framework does not take the computational overhead into account. In contrast, our method is based on a formalized tradeoff relationship between memory usage and the computational overhead and makes a search on a wider space of recomputation strategies than Chen's method.

We shall also mention that, at the same time as our publication, Kumar et al. [9] proposed a recomputation method based on pathwidth and treewidth. Lastly, while quite distant from our method as an algorithm, Bulatov recently posted on his blog[3] an informal idea to use tree decomposition to generate the recomputation strategies.

## 7   Conclusion

In this study, we proposed a new framework of recomputation method that can be applied to neural networks of any type and formulated the goal of the recomputation in the language of graph theory. Our framework enables much simpler treatment of the recomputation problem and possibly opens the door to complex methods with more sophisticated machinery of discrete mathematics. Also, in this study, we only considered a set of strategies that allows at most one recomputation per node. One possible future studies include the extension of our current formulation to strategies that allows multiple recomputations per node. While even more challenging, this future study may lead to even more efficient recomputation methods for neural networks.

We shall also note that we only considered static graphs in this study. One naive way to extend our algorithm to the dynamic setting is, for example, to conduct our algorithm in advance to the set of computation graphs that might become necessary in the course of training. In the case that the dimension of the input varies over the dataset, we may prepare a *maximal* graph to develop a computation strategy. Extension of our method to dynamic setting may open venues for new ways to optimize the training for heavy tasks like those in NLP and time series analysis.

## Acknowledgement

We thank Shinichiro Hamaji and Hiroto Imachi for technical suggestions.

## Footnotes

[2] For some networks, the approximate DP yielded slightly better results than the exact DP because the effect of the liveness analysis is not taken into the account in the definition of optimality in the general recomputation problem.

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
