[Supplementary Material · 692_supp.pdf]

## Appendix A  Pseudo-Code of Dynamic Programming Algorithm

We describe below the pseudo-code of the DP algorithm presented in Section 4. In the exact DP algorithm (Section 4.2), we need to set the input family of lower sets $\mathcal{L}$ to the set of all lower sets $\mathcal{L}_G$. In the approximate DP algorithm (Section 4.3), on the other hand, we need to set $\mathcal{L}$ to $\mathcal{L}_G^{\text{Pruned}}$.

We can obtain Memory centric strategy in Section 4.4 just by replacing $\min$ with $\max$ at line 15 in Algorithm 1.

---

**Algorithm 1:** Dynamic programming algorithm.

**input**  :Computational graph $G = (V, E)$, memory budget $B$, and family of lower sets $\mathcal{L}$
**output** :Increasing sequence of lower sets

```
# Init DP array
```
1  $\text{opt}[L, t] := \infty.$ $(L \in \mathcal{L}, t = 0, \ldots, T(V))$
2  $\text{opt}[\emptyset, 0] := 0.$
3  **for** $L \in \mathcal{L}$ *in ascending order of their set size* **do**
4    **for** $L' \in \mathcal{L}$ *s.t.* $L \subsetneqq L'$ *and* $t = 0, \ldots, T(V)$ **do**
5      $V' := L' \setminus L.$
6      $\mathcal{M} := \text{opt}[L, t] + 2M(V') + M(\delta^+(L') \setminus L') + M(\delta^-(\delta^+(L')) \setminus L').$
7      **if** $\mathcal{M} > B$ **then**
8        | **continue** # Memory constraint
9      $t' := t + T(V' \setminus \partial(L')).$
10      $m' := \text{opt}[L, t] + M(\partial(L') \setminus L).$
11      **if** $\text{opt}[L', t'] > m'$ **then**
          # Update DP array
12        | $\text{opt}[L', t'] := m'.$
13        | $\text{optarg}[L', t'] := (L, t).$
14 **if** *there exists* $t_0$ *such that* $\text{opt}[V, t_0] < \infty$ **then**
15    $t^* := \min\{t_0 \mid \text{opt}[V, t_0] < \infty\}.$
16    Compute increasing sequence $\{L_1 \prec \ldots \prec L_k\}$ by traversing $\text{optarg}$ from $(V, t^*)$ in reverse order.
17    **return** $\{L_1 \prec \ldots \prec L_k\}$
18 **else**
19    | **return** *Impossible*

---

## Appendix B  Configuration on Chen's Algorithm

In Chen's work, the procedure of topological-order in Algorithm 2 (Memory Optimized Gradient Graph Construction) and the definition of "candidate stage splitting points $C$" in Algorithm 3 (Memory Planning with Budget) are not clearly defined. In our experiments, we calculated the topological-order by performing DFS on the computation graph. Meanwhile, we define $C$ to be a set $v$ of nodes such that the removal of $v$ makes the graph disconnected, and calculated its value by enumerating the *articulation points* in the computation graph.

## Appendix C  Memory Consumption without Liveness Analysis

As an ablation study, we examined the memory consumption of both our algorithm and Chen's algorithm without liveness analysis. The result is shown in Table 2.

Our algorithm without liveness analysis worked much better than Chen's algorithm without liveness analysis. For example, our algorithm could reduce around 57% of memory in PSPNet, while Chen's algorithm reduced only 13%. Both Chen's algorithm and our algorithm, however, worked more poorly without liveness analysis than those with liveness analysis. Because we designed memory-centric strategy to be used with liveness analysis, the memory reduction with memory-centric strategy without liveness analysis was mediocre.

Table 2: The memory consumption without liveness analysis.

| Network | ApproxDP + MC | ApproxDP + TC | ExactDP + MC | ExactDP + TC | Chen's [2] | Vanilla |
|---|---|---|---|---|---|---|
| PSPNet | 3.2 GB (-57%) | 3.3 GB (-56%) | 3.3 GB (-56%) | 3.3 GB (-56%) | 7.6 GB (-13%) | 9.4 GB |
| U-Net | 6.8 GB (-21%) | 6.8 GB (-21%) | 6.8 GB (-21%) | 6.8 GB (-21%) | >=11.4 GB | 9.1 GB |
| ResNet50 | 4.7 GB (-42%) | 4.4 GB (-45%) | 4.7 GB (-42%) | 4.4 GB (-45%) | 6.7 GB (-22%) | 8.9 GB |
| ResNet152 | 2.8 GB (-61%) | 2.8 GB (-61%) | 2.8 GB (-61%) | 2.8 GB (-61%) | 4.1 GB (-48%) | 9.2 GB |
| VGG19 | 5.5 GB (-34%) | 5.5 GB (-34%) | 5.5 GB (-34%) | 5.5 GB (-34%) | 6.3 GB (-26%) | 7.0 GB |
| DenseNet161 | 1.9 GB (-70%) | 2.0 GB (-69%) | 1.9 GB (-70%) | 2.0 GB (-69%) | 3.4 GB (-55%) | 8.5 GB |
| GoogLeNet | 6.0 GB (-29%) | 6.0 GB (-29%) | 6.0 GB (-29%) | 6.0 GB (-29%) | >=11.4 GB | 8.5 GB |

For PSPNet, the approximate DP yielded slightly better results than the exact DP. This is because a small amount of memory that will be temporarily allocated during the computation of each node (e.g., convolutional node) is not taken into the account in our definition of the memory cost $M_v$ and thus such a memory allocation may slightly change the peak memory consumption from our expectation. The actual difference between "ApproxDP+MC" and "ExactDP+MC" of the peak memory consumption in PSPNet was 67 MB.

We observed that Chen's algorithm without liveness analysis worked worse than vanilla run in U-Net and GoogLeNet. This is because the vanilla run of Chainer conducts some local memory reduction by default. For example, for a pair of functions of the form "$h$=conv($x$); $a$=relu($h$)" at an activation layer of the network, the vanilla run of Chainer releases $h$ after the computation of $a$ if $h$ is not called from other functions. Meanwhile, Chen's algorithm without liveness analysis does not perform such a local-level optimization. Thus, it is possible that some recomputation algorithms without liveness analysis perform worse than the vanilla run with a local-level optimization for some network architectures.