[Reviews · NeurIPS 2019]

Reviewer 1



The paper proposes a method that reduces memory consumption of back prop, and as a result allows the use of larger batch sizes. The authors state that this is significant e.g. with batch norm, where batch size matters. In my own experience, I believe this is also significant for improved GPU utilization and data-parallel training. The paper is original in that I have not seen a similar treatment (although the actual solution may have been relatively straight-forward; asking the right question was an important part of this). The paper is well written and easy to follow. In Section 3, the paper formally defines the problem as a minimization problem under specified constraints, and then goes on to solve it in Section 4 via BP. I appreciate the systematic approach. The results are great, with in some benchmarks achieving an up to 50% further memory reduction compared to the baseline method (Chen's). I consider the validation of the method good and convincing. 4.4 "If the solution to the time-centric strategy does not exist, then try the memory-centric strategy next to prioritize memory reduction." I don't understand this. Time-centric refers to the optimal strategy. If that does not exist, that means a solution does not exist. Then why try the other strategy? Section 4.2 is very clear that DP is a correct solution to the problem. So please clarify 4.4.

Reviewer 2



The paper focuses on a broad problem that plagues many researchers today. The size of the models is often limited by the memory of the accelerator (GPU or TPU), the researcher is limited in what they can do. While there have been prior attempts to solve this problem in specific settings, they have been too limited to specific kinds of models and haven't seen broad usage or availability across ML tools and libraries. The paper does a good job of explaining the problem, coming up with a general solution that seems to apply to all kinds of directed graphs (most of deep learning), and empirically demonstrate the wins over existing strategies. While at first glance the submission seems incremental, the significance of it comes from the fact that it could now be more easily integrated into the underlying ML software and be made available to a lot more researchers.

Reviewer 3



First of all, I do believe that the general treatment of arbitrary computational graph is helpful. The authors build the constraints into the algorithm and uses a DP based algorithm to solve the planning under the given memory constraint. Given that the proposed algorithm generalize beyond what Chen’s algorithm can do, I would recommend the authors to include experiments on models that cannot be handled in Chen’s algorithm. This will help to strengthen paper. I want to point out that there are related treatment of using a tree decomposition https://medium.com/tensorflow/fitting-larger-networks-into-memory-583e3c758ff9 While I know we are not supposed to treat a blog post as existing literature since blogs are not peer-reviewed, the authors should still try to discuss it and give pointers to the related works. Finally, although this is a borderline paper, I do think it has some values and might be interesting to NeurIPS audiences. comment: I have changed the score to marginally above acceptance after reading authors feedback

[Author Response · NeurIPS 2019]

# Paper #692 Author Feedback

Thank you very much for the thorough reviews. We respond to each comments below.

**Responses to Reviewer #1:**

> 4.4 "If the solution to the time-centric strategy does not exist, then try the memory-centric strategy next to prioritize memory reduction." I don't understand this. Time-centric refers to the optimal strategy. If that does not exist, that means a solution does not exist. Then why try the other strategy?

It seems that we used a misleading phrasing here and we owe an apology. By the phrase "the solution of the time-centric strategy does not exist," we meant to refer to the case in which the time-centric strategy cannot satisfy the memory budget constraint *even after the application of liveness analysis* — we did not mean the case in which the DP solution does not exist. In such a case of memory shortage, what we suggest is to do the opposite of the objective and prioritize the memory; conduct the memory-centric strategy that maximizes the time, and implement the solution with liveness analysis. This compromise worked well in practice. We will rephrase this part in the revision.

> The runtime of the DP algorithm itself is not mentioned.

Please see Section 5.1; "The exact DP algorithm required more than 80 secs to complete for GoogLeNet and PSPNet, while the approximate DP completed within 1 sec for all networks."

> Is this feasible to be applied on the fly for every minibatch in a dynamic-network setting? For example, a Transformer MT model or lattice-free MMI in speech recognition, where each batch has different input/output lengths.

In this paper, we consider only static graphs. However, as future work, we may extend our algorithm to the dynamic setting by, for example, conducting our algorithm in advance to the set of computation graphs that might become necessary in the course of training. If the variable shape changes over the dataset, we may use maximum shape to develop a computation strategy. We will include this discussion as future work in the revised conclusion.

**Responses to Reviewer #2:**

> Would like to see some comparisons for sequence models (LSTMs) etc with the relevant work in that category.

If possible, we will try to include the additional experimental results in the revision.

> The directed graph approach works for many models, including "unrolled" sequence models, however for models including loop based sequences it may require some modifications to this approach. I believe it should still work though. The paper would be better if that was covered.

If the number of times the signal goes through each loop is fixed, we can unroll the loop by a simple manipulation on a computational graph. Then, we can apply our algorithm directly. As we mention in our response to Reviewer #1, it may be possible to modify our algorithm to extend the scope of applications. We plan to explore these modifications in future works further.

**Responses to Reviewer #3:**

> Given that the proposed algorithm generalize beyond what Chen's algorithm can do, I would recommend the authors to include experiments on models that cannot be handled in Chen's algorithm. This will help to strengthen paper.

As we show in the experiment, our algorithm greatly outperforms Chen's algorithm on PSPNet and U-Net in terms of memory consumption, and our method can reduce the computational overhead more greatly than Chen's algorithm when the memory resource is ample. Chen's algorithm is particularly not well-suited to U-Net; because of skip connections, there will always be a massive block in the decomposition of the computation graph. We plan to add a figure to visualize this explanation in the revision.

> I want to point out that there are related treatment of using a tree decomposition https://medium.com/tensorflow/fitting-larger-networks-into-memory-583e3c758ff9 While I know we are not supposed to treat a blog post as existing literature since blogs are not peer-reviewed, the authors should still try to discuss it and give pointers to the related works.

We will make a pointer to the blog-post as an example of the implementation of Chen's algorithm and mention its ideas. We would humbly like to ask the reviewer, however, to recognize that our work is the first work in the community to investigate the algorithm applicable to general graph with appropriate formality and to experimentally verify its efficacy. We also plan to publish the implementation upon the publication.

[Meta-Review · NeurIPS 2019]

This work formalizes the problem of minimizing memory consumption through recomputation when performing a forward-backprop evaluation of a computation graph; it provides an optimal dynamic programming algorithm and an efficient heuristic and demonstrates strong improvements in memory savings over existing methods. Three expert reviewers initially assess the paper as 8/8/5, and the authors provided a detailed rebuttal. All reviewers took part in a discussion and the final assessment is 8/8/6, with reviewer consensus that this method is practically useful and the reported gains are strong. Overall this work is a nice contribution.